5

# High-Resolution Virtual Catchment Simulations of the Subsurface-Land Surface-Atmosphere System

Bernd Schalge<sup>1</sup>, Jehan Rihani<sup>2</sup>, Gabriele Baroni<sup>3,4</sup>, Daniel Erdal<sup>5</sup>, Gernot Geppert<sup>6</sup>, Vincent Haefliger<sup>1</sup>, Barbara Haese<sup>7</sup>, Pablo Saavedra<sup>1</sup>, Insa Neuweiler<sup>8</sup>, Harrie-Jan Hendricks Franssen<sup>9,11</sup>, Felix Ament<sup>6</sup>, Sabine Attinger<sup>3</sup>, Olaf A. Cirpka<sup>5</sup>, Stefan Kollet<sup>9,11</sup>, Harald Kunstmann<sup>7,10</sup>, Harry Vereecken<sup>9,11</sup>, Clemens Simmer<sup>1</sup>

<sup>1</sup>University of Bonn, Meteorological Institute, Bonn, Germany <sup>2</sup>University of Cologne, Mathematical Institute, Cologne, Germany <sup>3</sup>Helmholtz-Center for Environmental Research, Leipzig, Germany

<sup>4</sup>University of Potsdam, Institute of Earth and Environmental Science, Potsdam, Germany
 <sup>5</sup>University of Tuebingen, Center for Applied Geoscience, Tuebingen, Germany
 <sup>6</sup>University of Hamburg, Meteorological Institute, Hamburg, Germany
 <sup>7</sup>University of Augsburg, Institut fuer Geographie, Augsburg, Germany
 <sup>8</sup>Leibniz University Hannover, Institut fuer Stroemungsmechanik und Umweltphysik im Bauwesen, Hannover, Germany

<sup>9</sup>Forschungszentrum Juelich GmbH, Agrosphere (IBG-3), Juelich, Germany <sup>10</sup>Karlsruhe Institute of Technology (KIT), Institute of Meteorology and Climate Research (IMK-IFU), Garmish-Partenkirchen, Germany <sup>11</sup>Centre for High-Performance Scientific Computing (HPSC-TerrSys), Geoverbund ABC/J, Juelich, Germany

Correspondence to: Bernd Schalge(bschalge@uni-bonn.de)

- **Abstract.** Combining numerical models, which simulate water and energy fluxes in the subsurface-land surface-atmosphere system in a physically consistent way, becomes increasingly important to understand and study fluxes at compartmental boundaries and interdependencies of states across these boundaries. Complete state evolutions generated by such models, when run at highest possible resolutions while incorporating as many processes as attainable, may be regarded as a proxy of the real world a virtual reality which can be used to test hypotheses on functioning of the coupled terrestrial system and may serve
- as source for virtual measurements to develop data-assimilation methods. Such simulation systems, however, face severe problems caused by the vastly different scales of the processes acting in the compartments of the terrestrial system. The present study is motivated by the development of cross-compartmental data-assimilation methods, which face the difficulty of data scarcity in the subsurface when applied to real data. With appropriate and realistic measurement operators, the virtual reality not only allows taking virtual observations in any part of the terrestrial system at any density, thus overcoming data-scarcity
- problems of real-world applications, but also provides full information about true states and parameters aimed to be reconstructed from the measurements by data assimilation. In the present study, we have used the Terrestrial Systems Modeling Platform TerrSysMP, which couples the meteorological model COSMO, the land-surface model CLM, and the subsurface model ParFlow, to set up the virtual reality for a regional terrestrial system roughly oriented at the Neckar catchment in southwest Germany. We find that the virtual reality is in many aspects quite close to real observations of the catchment
- concerning, e.g., atmospheric boundary-layer height, precipitation, and runoff. But also discrepancies become apparent both

in the ability of such models to correctly simulate some processes – which still need improvement - and the realism of the results of some observation operators like the SMOS and SMAP sensors, when faced with model states. In a succeeding step, we will use the virtual reality to generate observations in all compartments of the system for coupled data assimilation. The data assimilation will rely on a coarsened and simplified version of the model system.

#### 5 1 Introduction

30

Numerical models, which simulate water and energy fluxes in the subsurface-land surface-atmosphere system in a physically consistent way, are important ingredients to climate and weather prediction, flood forecasting, water resources management, agriculture, and water quality control (Shrestha et al. 2014; Larsen et al. 2014; Simmer et al., 2015). Recent developments embrace the combination of coupled land surface-subsurface models with data assimilation (DA) techniques in order to obtain

- real-time forecasts of the development of the coupled system (Shi et al., 2014, 2015; Ridler et al., 2014; Kurtz et al., 2016). Different data assimilation philosophies and methods exist encompassing nudging, variational and sequential ensemble-based methods (e.g. Kalnay 2003). Yet the development and choice of appropriate data assimilation methods is limited by the availability of observational information across all compartments as well as by appropriately characterized real-world systems. The use of virtual reality (VR) systems that provide a virtual truth and deliver full information about the subsurface-land
- surface-atmosphere system can be key for the development of a consistent data assimilation framework. Virtual realities have already been used for specific compartments of the coupled terrestrial system (see Fatichi et al., 2016 and reference herein). Bashford et al. (2002) computed virtual remote-sensing observations with 1 km resolution to derive, among others, process parameterizations for evapotranspiration in a hydrological model operating on the same scale as the remote sensing data. Weiler and McDonnel (2004) used a virtual-reality approach on the hill-slope scale to detect and quantify
- the major controls on subsurface flow processes and derive tunable parameters for conceptual models. Virtual experiments were defined by Schlueter et al. (2012) to explore the relationship between soil architecture and hydraulic behavior and by Chaney et al. (2015) for testing sampling designs. In groundwater hydrology it is a standard procedure to test inverse modeling and data assimilation approaches on virtual computational aquifers (e.g., Zimmermann et al., 1998; Hendricks Franssen et al., 2009). These are used to generate realistic data of an aquifer with exactly known hydraulic and geochemical properties at every
- point (e.g., Schaefer et al. 2002). More recently also integrated modeling approaches were considered to generate virtual realities (Mackay et al., 2015).

The application of the virtual-reality concept for developing data-assimilation systems has the following main advantages:

(I) For the prediction of fluxes and states in a real-world subsurface-land surface-atmosphere system, we do not know whether the errors in our predictions are due to incomplete model specification (model structural errors), wrong parameterizations, erroneous model parameters, measurement errors, or defects in data assimilation. Working with a virtual

truth allows separating model structural error both in the coupled terrestrial model and in the measurement operators from the

proper data-assimilation problems, like the generation of the ensemble or the choice of the filters. Working only with realworld data would hardly allow the attribution of prediction errors to their causative sources.

(II) The immense scale complexity of terrestrial systems renders already the estimation of its state often very challenging despite the wealth of observations available nowadays from in-situ and remote sensing. Available measurements never cover

- all locations and times, thus only parts of the heterogeneities can be probed. Moreover, any measurement technology sees only a certain part of reality. It is not always clear whether observations contain enough information with the appropriate spatial and temporal resolution to substantially improve predictions. With the virtual reality we may circumvent data scarcity in dataassimilation problems. In real-world applications, there are never enough observations available to define a "truth" against which data-assimilation methods can be tested. The virtual-reality approach allows generating arbitrarily long time series and
- high-density observations to work with, and it facilitates comparing estimated states and parameters to the truth everywhere at all times. This is especially true for coupled subsurface-land surface-atmosphere problems, for which data assimilation has not been evaluated and tested until now.

(III) Studies on data assimilation using different observation densities, observation errors, and data types will provide insight into data value and thus contribute to the development of better monitoring designs for improved predictions of states and fluxes.

and fluxes.

(IV) Virtual realities are a unique testbed to develop and validate data-assimilation methods as well as the impact of the underlying assumptions (e.g., Gaussianity). In particular, weakly and fully coupled data assimilation can be compared. In weakly coupled data assimilation, sequential data assimilation is done for multiple compartments of the terrestrial system (e.g., groundwater, land surface and atmosphere), but observations made for a given compartment only update states (and possibly

parameters) within that compartment and not in other compartments. In fully coupled data assimilation, by contrast, observations are available for multiple compartments of the terrestrial system and are used to update states (and possibly parameters) across different compartments of the terrestrial system through cross-correlations, which may be inferred from ensemble statistics of the fully coupled system.

(V) Finally, virtual realities allow examining the effects of conceptual assumptions regarding process descriptions of
 various degrees of complexity, spatial resolution of heterogeneities, and representation of boundary conditions, among others, on overall system behavior.

In this paper we present the development and testing of a catchment-based virtual-reality system as a first step in the development of a weakly or strongly coupled data-assimilation system; such systems allow for the assimilation of all observation types made in all compartments of the subsurface-land surface-atmosphere system at the same time. Our virtual

reality represents the fully coupled terrestrial system from the bedrock to the upper atmosphere covering the catchment of a higher-order river (length  $\approx$  380km, area  $\approx$  14000km<sup>2</sup>) with a buffer zone surrounding it, in which we simulate - as realistically as possible - the evolution of states in all compartments and the water and energy fluxes throughout the system. This virtual reality provides a virtual truth, from which arbitrary (virtual) observations can be made and which avoids the lack of information for a thorough evaluation of data-assimilation systems. Towards this ends, the model of the subsurface-land

surface-atmosphere system needs to represent the strong spatial variability especially of the land component, which affect the overall system behavior due to nonlinear couplings and feedbacks.

Setting up and executing a fully coupled virtual reality comes with challenges. Without a sufficiently detailed representation of the relations and dependencies of fluxes, states, and parameters in different compartments, the applicability of any dataassimilation method derived from the setup of the virtual reality will be questionable. Since - for a chosen size of the catchment

- 5 - IT-requirements limit the spatial resolution of the fully coupled model, a balance needs to be found between the detail in process representation and computational feasibility. Accordingly, for some processes appropriate upscaling rules or apparent parameters need to be introduced and applied in order to guarantee that the interrelations between different observations and forcings are adequately represented. Here we report on first steps to derive the virtual reality for a catchment and discuss in
- 10 particular the limitations.

Since a virtual reality with no resemblance to a real world catchment hardly allows for evaluating its realism, we base our virtual reality loosely on the Neckar catchment in southwest Germany, that contains variable topography, different land cover, high and low precipitation regions, deep and low water tables, regions prone to flooding events etc. (see Figure 1). The model does not aim at exactly reproducing the catchment's response to hydro-climatic forcing, instead we only require that the

- simulated response is realistic with respect to typical spatial and temporal characteristics. The model needs to simulate realistic 15 responses in all state variables forming the basis for typical measurements eventually produced by measurement (forward) operators. This requires that the virtual reality is run at very high resolution. Thus, the main objective of this paper is to explain how we created the virtual reality and prove the realism of the generated data and observations.
- The remainder of the paper is structured as follows. In section 2, we introduce the simulation platform TerrSysMP, while 20 Section 3 describes in detail the surface and subsurface parameters for topography, soil, land use, vegetation, and the river network. In Section 4, we show some snapshots and time series of state variables or system parameters extracted from the virtual reality. In Section 5, we compare virtual observations obtained from the virtual reality with real observations to demonstrate how the most important requirements for the virtual reality are met. These results are discussed in Section 6 together with several issues, which came up when developing the virtual reality. We provide conclusions and an outlook in
- Section 7. 25

## 2 The Terrestrial Systems Modeling Platform (TerrSysMP)

We used the Terrestrial System Modeling Platform (TerrSysMP, see Shrestha et al. 2014; Gasper et al. 2014; Sulis et al. 2015) developed within the Transregional Collaborative Research Centre TR32 (see Simmer et al. 2015) for the generation of the virtual reality (VR). TerrSysMP couples (Figure 2) the hydrologic flow model ParFlow v693 (Ashby and Falgout, 1996; Jones

and Woodward, 2001; Kollet and Maxwell, 2006), the Community Land Model, CLM v3.5 (Oleson et al, 2008), and the 30 Consortium for Small Scale Modeling (COSMO v4.21) model (Baldauf et al. 2011) via the Ocean Atmosphere Sea Ice

Coupling framework OASIS3 (e.g. Valcke et al., 2006), using a dynamical two-way approach including down- and upscaling algorithms for fluxes and state variables between computational grids of different resolution.

ParFlow is a variably saturated watershed flow model which solves the three-dimensional Richard's equation to model saturated and unsaturated flow in the subsurface, and the fully integrated kinematic wave equation to model two-dimensional

- 5 overland flow. Other global and regional hydrological models use this equation to route overland flow, being already tested in the MODCOU model (Haefliger et al. 2015) and currently used in the TRIP model (Alkama et al. 2012). Advanced Newton-Krylov multigrid solvers are used, that are especially suitable for massively parallel computer environments. Excellent model performance and parallel efficiency have been documented by Jones and Woodward (2001), Kollet and Maxwell (2006), and Kollet et al. (2010). A unique feature of ParFlow is the use of an advanced octree data structure for rendering overlapping
- objects in 3-D space, which facilitates modeling of complex geology and heterogeneity as well as the representation of topography based on digital elevation models and watershed boundaries.

CLM is a single column biogeophysical land-surface model considering coupled snow-, soil-, and vegetation-processes. CLM is released by the National Center for Atmospheric Research (NCAR). Land surface heterogeneity is represented as a nested subgrid hierarchy in which grid cells are composed of multiple land units (glacier, lake, wetland, urban, and vegetation),

- 15 snow/soil columns (to capture variability in snow and soil state variables within each land unit), and Plant Functional Types (PFTs) to capture the biogeophysical and biogeochemical differences between broad categories of plants in terms of their functional characteristics. In TerrSysMP, the 1-D Richards-equation model included in CLM is replaced by ParFlow. COSMO is a limited-area, non-hydrostatic numerical weather prediction model, which operationally runs at the German
- weather service DWD, among others, for Numerical Weather Prediction (NWP) and various scientific applications on the 20 meso-β and meso-γ scale. COSMO is based on the primitive thermo-hydrodynamical equations describing compressible flow in a moist atmosphere. As a limited-area model, COSMO needs lateral boundary conditions from a driving larger-scale model. We impose the lateral conditions by nesting COSMO into COSMO-EU which spans all of Europe, or in COSMO-DE which spans Germany. The lateral boundary formulation used is a relaxation technique in which the internal model solution is nudged against an externally specified solution over a narrow transition zone between the two domains.
- 25 Within OASIS3, the upscaling algorithm uses the mosaic or explicit sub-grid approach (Avissar and Pielke 1989) in which high-resolution land surface fluxes are averaged to be transferred to the coarser resolution of the atmospheric model component. The implemented Schomburg scheme (Schomburg et al., 2010, 2012) downscales atmospheric variables of the lowest atmospheric model layer to the higher-resolved land surface model. The scheme involves (i) spline interpolation while conserving mean and lateral gradients of the coarse field, (ii) deterministic downscaling rules to exploit empirical relationships
- 30 between atmospheric variables and surface variables, and (iii) the addition of high-resolution variability (i.e. noise) in order to restore spatial variability. The scheme has been developed and tested for downscaling 2.8 km atmospheric simulations by COSMO to 400 m for the land surface.

TerrSysMP allows simulating the terrestrial water, energy, and biogeochemical cycles from the deeper subsurface including groundwater (ParFlow) across the land-surface (CLM) into the atmosphere (COSMO). Water and energy cycles are coupled

5

via evaporation and plant transpiration; these processes are modeled by CLM with a non-linear coupling to ParFlow through soil-water availability and root-water uptake (Figure 2). The two-way coupling between CLM and COSMO encompasses radiation exchange and turbulent exchanges of moisture, energy, and momentum. OASIS3 allows different temporal and spatial resolutions of the coupled model components. For example, a temporal resolution of 15 minutes is sufficient for the subsurface and land-surface components, whereas time steps as small as 5 seconds are needed for the atmosphere. A higher spatial resolution can be assigned for the surface and subsurface parts to allow for a better representation of soil and land-use heterogeneity. Each model component can also be used standalone; coupled runs of ParFlow and CLM using atmospheric inputs from offline driven COSMO simulations or other sources are as well possible as COSMO runs coupled with CLM without ParFlow, and thus without 3-D soil and groundwater dynamics.

#### 10 3 Description of the Virtual Reality

Our virtual reality is based on the Neckar catchment in the south-west German state of Baden-Wuerttemberg (see Figure 1), east of the mountain range of the Black Forest and north of the Jurassic ridge of the Swabian Alb. Some features that are characteristic for the domain, such as middle Triassic and Jurassic karst aquifers, are not included to avoid the manifold hydrological challenges related to its modeling. The catchment has a varying topography including mountains up to 1050 m

a.s.l., river valleys, different land use types, i.e. grassland, cropland (majority of the area), broadleaf and needle leaf forest (see Figure 3), and soil classes. The computational domain of the virtual reality is larger than the Neckar catchment to allow the atmospheric model to develop its own internal dynamics.

Annual mean precipitation over the real catchment ranges between 600 and 2000mm (see Section 3.1) with highest values over the Black Forest. Inter-annual variability of precipitation can deviate by up to one third of the mean value. Monthly

- precipitation can vary largely and its mean annual cycle is weak with slightly lower values in spring and autumn. While summer precipitation is dominated by convection, winter precipitation is predominantly related to fronts of extra-tropical cyclones with enhanced precipitation over the mountains due to orographic lift. Daily average temperatures vary with altitude between -5°C and 0°C in January and 13 and 18°C in July. Land use and cover in the lower elevations are dominated by agriculture while the Black Forest features mainly needle-leaf trees. Broad-leaf trees can be found over smaller areas throughout the catchment.
- Distance to groundwater is in large parts of the area restricted to a few meters, which assures strong coupling between groundwater depth and evapotranspiration.

The described features are quite typical for central Europe, which led us to choose the Neckar catchment as the location of the virtual reality. We use two realizations with different spatial extensions, spatial resolutions and coupling degrees as intermediate steps towards a fully coupled system. Domain 1 covers the entire area at a somewhat coarser resolution, whereas

the more finely resolved Domain 2 is restricted to the upper Neckar catchment, defined by the gaging station Plochingen. This two-step approach better allows experimenting with the model and evaluating its realism. In addition, there is currently a software restriction, which does not allow for cases with more than 4.2 million CLM columns; thus currently the high resolution

of Domain 2 cannot be used for the full Neckar catchment area of Domain 1. Domain 1 is a rectangular area of ~57,850km<sup>2</sup> which encompasses the whole Neckar catchment of ~14,000km<sup>2</sup>. Coupling is restricted to COSMO and CLM, i.e. the 3-D groundwater flow representation by ParFlow is deactivated and the spatial resolution is reduced compared to Domain 2. The latter covers only the upper Neckar catchment (2,621km<sup>2</sup>) and restricts coupling to CLM and ParFlow with COSMO output used as atmospheric forcing (Figure 1).

5

#### 3.1 Domain 1: Entire Neckar Catchment with Bounding Areas

In Domain 1, both COSMO and CLM run at the same horizontal resolution of ~1.1km (0.01° in rotated COSMO latitudelongitude coordinates). Due to projections between real (CLM) and rotated (COSMO) geographical coordinates, misplacements of land surface pixels in CLM in relation to their original locations can be up to half a grid diameter.

- COSMO is set up identical to the operational COSMO-DE setup of the German Weather Service DWD, e.g., the deep 10 convection parameterization is switched off because at the chosen grid resolution convection is enabled by the dynamical core (see Section 2.1). In COSMO-DE, the operational solution is 2.8km, so that the approximation regarding deep convection is even more appropriate in our virtual reality. Similar choices were taken by Smith et al. (2015), who simulated precipitation events of roughly the same domain using nested WRF models, where cumulus parameterization was switched off at horizontal
- 15 resolutions of 900m and 300m. Lateral boundary forcing and constant fields (topography, land-mask etc.) are provided by the COSMO-DE analysis fields, which are downscaled to the 1.1km grid by linear interpolation. The lateral relaxation zone, which moderates the jump from the lateral driving fields to the inner model area, is set to 12km.

For setting up CLM, the European digital elevation model (DEM) by the European Environment Agency EEA (http://www.eea.europa.eu/data-and-maps/data/eu-dem) was projected to the latitude/longitude grid and bi-linearly

- interpolated to 1.1km from the original 30m spatial resolution. Land use is taken from the 2006 Corine Land Cover Data Set 20 (http://www.eea.europa.eu/data-and-maps/data/corine-land-cover-2006-raster-3) and also provided by EEA. Since the latter dataset features many more land use types (at a resolution of 100m) than required by CLM, they were grouped according to the CLM (IGBP) Plant Functional Type classes (1) broad-leaf forests, (2) needle-leaf forests, (3) grassland, (4) cropland, and (5) bare soil. Urban areas are not considered in our virtual reality and replaced by bare soil. Water surfaces (e.g., larger lakes
- 25 like Lake Constance in the South of Domain 1) are also treated as (always fully saturated) bare-soil in CLM while COSMO uses its own land-mask and specific calculations for water surfaces. Therefore, no values from CLM are used for water surfaces. A few hundred grid cells feature shrubs (mostly areas that are re- or de-forested or areas at higher altitudes) which are treated as forests, and each grid cell features only one - the most dominant - plant functional type. The plant Leaf Area Index (LAI) is computed from MODIS (Myneni et al. 2002) as monthly averages for the year 2008 for each of the four
- vegetated land use classes. This LAI is increased for all plant functional types by about 20 percent in the summer months and 30 significantly changed from factors less than 1 to about 3.3 in winter-time for needle-leaf forests in order to account for known biases in the MODIS data (Tian et al. 2004). The stem area index (SAI) is estimated from the LAI by a slightly modified (no dead leaf for crops, constant base SAI of 10 percent of maximum LAI) formulation of Lawrence and Chase (2007) and Zeng

et al. (2002) to better represent European tree types. Vegetation height was set to 7m for needle-leaf trees, 10m for broad-leaf trees to account for partial coverage by shrubs, and to 20-120cm for crops and 10-60 cm for grass depending on the time of the year, with low values in the winter months and largest values in July and August. Since we consider only one crop type, we do not specify a harvest date when the plant height drops to its minimum, but assume a smooth decline between August and October.

 For the representation of soils in CLM we use the 1:1,000,000 soil map (BUEK1000) provided by the Federal Institute for

 Geosciences
 and
 Natural
 Resources
 BGR

 (http://www.bgr.bund.de/DE/Themen/Boden/Informationsgrundlagen/Bodenkundliche\_Karten\_Datenbanken/BUEK1000/bu
 ek1000\_node.html)

- This soil map is available for entire Germany; thus only small areas in Switzerland and France are missing outside the Neckar catchment for which we assume a nearby soil class. BUEK1000 offers sand and clay percentages as well as carbon content for two to seven soil horizons down to a maximum depth of 3m for each soil type, which we transferred to the respective model layers by choosing the horizon found at the center of each model layer. For lower soil layers without BUEK1000 information the lowest horizon is assumed to extend down to the 3m required for CLM. Carbon content is used to infer soil color. For
- urban areas (modeled as bare soil, as mentioned above) a fixed soil color (class 8 in CLM) was used different from land use classes with plants.

Simulations for Domain 1 were performed for a period of seven years (2007-2013). A 100-day spin-up (each day forced with atmospheric data from 01/01/2007) for CLM was performed prior to the simulation start, which created the virtual reality.

# 3.2 Domain 2: Upper Neckar Catchment

- The upper Neckar catchment simulations with CLM-ParFlow were performed with 100m horizontal resolution for both submodels. The upper 10 layers of ParFlow and CLM are identical, while 40 layers were added in ParFlow down to 100 m depth to account for groundwater dynamics. Different from the rectangular Domain 1, Domain 2 contains only grid points within the Upper Neckar catchment, which reduces the computational and storage demand (compared to an encompassing rectangle domain) by about 50%. The lowest atmospheric layer state variables simulated for Domain 1 are spline-interpolated to 100m
- and used as atmospheric forcing.

Since soil properties may vary substantially at scales smaller than the 1km for which BUEK1000 is appropriate, which might impact system dynamics (Binley et al. 1989, Herbst et al. 2006, Rawls 1983), the soil map of Domain 1 is downscaled by artificially adding variability using the following method:

The BUEK1000 soil map is randomly sampled at 1995 point locations with one sample every 5 km2 on average, a
 minimum sample distance of 250 m, and at least one sample for each soil type of the original soil map. This strategy resulted from extensive testing by minimizing the tradeoffs between reproducing the main features of the original soil map and creating variability at finer resolution.

(1) The sample locations are used as conditional points for further interpolation. Here, texture, carbon content and depth of the first three soil horizons are extracted from the BUEK1000.

(2) Experimental variograms and cross-variograms are calculated for all variables and exponential models were fitted to all spatial structures.

(3) Horizon depths and carbon content are interpolated based on ordinary kriging. Texture map (sand and clay percentage) is generated using a single realization based on conditional co-simulation (Gomez-Hernandez and Journal, 1993) to provide the sub-scale variability (<1 km2).

(4) Since ParFlow describes retention and hydraulic conductivity curves based on van-Genuchten-Mualem parameters, pedotransfer functions are applied to estimate these parameters. The pedotransfer functions of Cosby et al. (1984), Rawls

(1983) and Tóth et al. (2015) are used and selected based on data availability, applicability of the particular approaches, and previous evaluations conducted in the area (Tietje and Hennings, 1996).

Apart from the described changes in the soil map, the CLM setup is identical to the one for Domain 1. In order to keep soil porosity identical between CLM and ParFlow, we replaced the porosity calculation within CLM (which uses a different pedotransfer function). The Manning's surface roughness was set to a constant of 5.52×10-4 h/m1/3 and the specific storage

to 1×10-3. The chosen surface roughness value results in a realistic base flow for the local rivers without calibration. Repercussions of this choice are discussed in Chapter 6. Slopes in rivers are modified/smoothed to avoid artificial ponded areas.

In order to allow for realistic flow in the saturated zone, a 3-D geologic of the geological survey of the state of Baden-Wuerttemberg was used from which eleven rock types for Baden-Wuerttemberg were identified (see Figure 4). Nine of these

20 types can be found in the upper Neckar catchment albeit some only cover very small volumes. Tab. 1 summarizes porosity and hydraulic conductivity used in the virtual reality for the different stratigraphic units. It should be noted that karst features of limestones are not considered at all, and porosities in stratigraphic units containing limestones and crystalline rocks are generally chosen considerably higher than in nature.

Not covered in any of the data sets discussed and implemented so far are the large alluvial bodies filling large part of the

- Neckar valley throughout the domain (Riva et al. 2006). As these bodies are highly conductive, they are also believed to be important for the regional groundwater flow and should be included in the model. Since the valleys in the model are often small compared to the limited horizontal resolution of the model, we conceptualize the alluvial bodies as gravel layers existing underneath all river cells (cells with a pressure head >0.1) and directly next to rivers (riverbanks) (i.e., one gridpoint besides each river cell). The gravel layers reach from beneath the soil down to a depth of 8m. The gravel cells are parameterized with
- a high hydraulic conductivity of 1 m/h, a porosity of 0.6 and van-Genuchten parameters of 2 for n and 4 for  $\alpha$  [1/m] (residual saturation is 0.06 cm3/cm3). In reality, up to 30% of the runoff takes place in the subsurface, especially during periods of base flow. This is based on a calculation of the upper Neckar sub-catchment where we used measured gridded precipitation data as well as measured river discharge data at the outlet, combined with the virtual-reality data for evapotranspiration to calculate the water balance over a whole year to eliminate any influence of transient storage changes. This is possible as we used the

year 2007 which was also used to perform the spin-up resulting in negligible storage difference between the years. About 30% of the precipitation was neither evaporated nor discharged by the river. While our simulated evapotranspiration rates may be inaccurate, it is implausible to assume this can account for 30% of the precipitation. The only other method how the water could have left the domain is through the subsurface. Thus, gravel channels are needed to account for this lateral flow, as

- otherwise subsurface outflow from the domain would be negligible. The gravel layers below the rivers allow for a reasonable distribution of surface and subsurface discharge at the outlet of the catchment and reasonable river aquifer exchange fluxes. It should be noted that the model assumes no-flow boundary conditions at the subsurface domain boundaries. A very conductive (100m/h) layer close to the model outlet was included to enable the subsurface water to discharge back into the river and leave the model domain as the no-flux boundary condition forces the water to rise to the surface in order to leave the
- domain. To avoid influence of this slightly artificial boundary, all observations considered are taken at a large enough distance from the outlet boundary.

#### 4 Demonstration Results

In the following, we present some example results of the virtual-reality simulations in order to demonstrate its potential for a better understanding of the dynamics in coupled terrestrial systems and to show that it resembles reality well enough to serve

- in the test of data-assimilation strategies. Figure 5 shows a snapshot of Domain 1 showing the three-dimensional distribution of cloud water/ice (greyscale), precipitation density (blue, foreground over grey clouds) as well as the volumetric soil moisture (colored). The soil shows different layers of soil moisture where changes are mainly connected to different soil hydraulic properties. Only clouds reaching high enough to have sufficient cloud ice p