# Peer review of "High-Resolution Virtual Catchment Simulations of the Subsurface-Land Surface-Atmosphere System"

_Hydrology and Earth System Sciences, 2016_

## Referee Comment (RC1) · Anonymous Referee #1 · 29 Nov 2016

This manuscript describes the numerical generation of a virtual reality (VR) of a subsurface-land surface – atmosphere system. The model system applied for generating the VR is the TerrSysMP platform coupling the COCMO meteorology, CLM land surface and ParFlow subsurface model. The required data are obtained from the Neckar catchment, however significant physical characteristics describing the watershed, such as the karstic properties of the limestone areas, are neglected here for simplicity. The generated VR is then compared to the boundary layer height, precipitation, and runoff measurements, as well as to spatially distributed soil moisture information from SMOS and SMAP sensors. The structure of the paper is rather clear as is the intention of the authors; they wish to provide a first reference publication in order to

then proceed to use this VR for future data assimilation exercises. However, while the structure and intention of the authors is clear, I have asked myself while reading the manuscript and preparing the review; "What have I actually learned from the paper?":

- The modelling platform and individual components have been used previously and were also tested and compared against real data elsewhere, so there are no general new insights, except maybe that is has not yet been done explicitly for the Neckar region before.

- The difficulties in relating microwave data to land surface soil moisture conditions is also well known and widely published.

- The dependency of ET to soil moisture availability and water table depth as outlined in section 4 is, in my opinion, basic soil physics material that is taught in every introductory course.

- The only surprising point for me, is how well the yearly precipitation amounts are actually covered.

In summary, I see the principle need and the desire for the authors to prepare for their next steps. However, I do not think that the current version of manuscript provides sufficient and substantial new information to potential readers to justify it as a "stand-alone" paper in HESS. I could anticipate some content of this manuscript in a very condensed form (and without losing any of the provided information) as a technical part of future more data-assimilation type papers. Therefore, in the present form I would suggest to reject the paper for publication in HESS.

---

## Author Comment (AC1) · 7 Dec 2016

Dear Reviewer,

thank you for your feedback and suggestions made regarding our paper. We would like to highlight the two major motivations for this paper. The first is to provide insights into the problems we faced while constructing the VR and how to overcome them. We believe this material is valuable for scientists working on similar problems because our lessons learned are not particular to our specific problem but are of general value to the modelling and the Data Assimilation (DA) community. We agree the manuscript lacks some emphasis on this part and we will remedy this in the modified version.

Second, the VR created is designed as a basis for data assimilation (DA) experiments. Similar efforts are often criticized for insufficient proof that the synthetic observations created are comparable to real observations. We demonstrate that due to the high resolution and the number of processes involved in the coupled approach, which puts the size of the domain at the edge what is currently possible, our VR can be used for DA experiments. We show that TerrSysMP produces reasonable results in all compartments with the expected variability. We discuss insufficient results concerning passive microwave satellite observations and some aspects of river discharge, and we explain how we deal with these problems.

In the following we respond to your more specific comments.

'The modelling platform and individual components have been used previously and were also tested and compared against real data elsewhere, so there are no general new insights, except maybe that is has not yet been done explicitly for the Neckar region before.'

While the TerrSysMP model system has been evaluated to some degree before, it remains state-of-the-art and we highlight in the paper the aspects in this model which are most important for DA experiments. Amongst those are discharge comparisons with real observations of the uncalibrated model system. Our results show the model is able to produce good results without calibration, and we discuss how the simulation can be improved in the future. Houtekamper et al. (2004) state that e.g. for atmospheric DA, questionable results might ensue from using virtual observations given "poorly understood model imperfections". Coupled models such as TerrSysMP are even more complex than the models used then and thus need thorough tests before being used as a VR in DA experiments. Our results demonstrate this and are designed to be used as a benchmark for the VR concept. Existing model studies on the Neckar catchment use simpler, mostly uncoupled models, which are specifically designed to reproduce certain model outputs when calibrated, river discharge for instance (e.g., Samaniego et al. 2010). Such models, however, cannot be used for cross-compartmental DA

which requires coupled simulation of all processes with potential to be sensed by monitoring systems. We acknowledge the shortcoming in the reproduction of the timing of discharge peaks by TerrSysMP, which however would be similar for all models using homogeneous grids for representing river flow, but also suggest a remedy as outlined in the paper. In a revised version, we would make this point clearer and put more focus on this point.

'The difficulties in relating microwave data to land surface soil moisture conditions is also well known and widely published.'

While extensive literature exists on satellite remote sensing of soil moisture, little has been done regarding the effect of resolution when soil moisture is used as a variable for data assimilation. Our study paves the way to evaluate such and other effects. While existing parameterizations for L-band, such as CMEM in our case, are well suited for SMOS retrievals, we find large differences in brightness temperature distributions between our VR and SMOS/SMAP observations. We hypothesize that these differences mainly originate from CMEM parametrization defaults oriented at the ECO-CLIM/TESSEL land model, and are not suitable for the CLM included in TerrSysMP. In a revised version we will modify this section accordingly and present further ideas to ameliorate these problems.

'The dependency of ET to soil moisture availability and water table depth as outlined in section 4 is, in my opinion, basic soil physics material that is taught in every introductory course.'

The aim of the data assimilation tests will be to evaluate the effect in cross-compartments (e.g. the impact of assimilation of groundwater heads to the simulated evapotranspiration). For this reason, ET-water table relation is a very important hydrological process that should be reproduced in the VR. Consequently, we were quite happy that our VR exhibits the expected relation. In a revised version of the paper we will significantly shorten this section and put more weight on the inter-annual soil

moisture dynamics of the catchment and address the low inter-annual variability and wet bias including strategies to solve these problems. Nonetheless, we do see the importance of highlighting the issues which arise from using physically-based models rather than extensive calibration to achieve expected model dynamics. Additionally, we would focus more on the added value of high resolution by including comparisons of fluxes and state variables in the overlap area of the two domains.

We agree, more emphasis is needed on the 'lessons learned' part of the paper. We will point out some of the novel aspects more clearly, while the validation sections will not suffer much from significant shortening to warrant a publication as a stand-alone paper. Currently we are running the fully coupled system (atmosphere at 1.1 km, surface and subsurface at 400 m) and plan to present initial results for comparison in a revised version. This is particularly interesting regarding the comparison of fluxes in the domain overlap mentioned above.

Kind regards,

the authors

References:

Houtekamer, P. L., Mitchell, H. L., Pellerin, G., Buehner, M., Charron, M., Spacek, L., & Hansen, B. (2005). Atmospheric data assimilation with an ensemble Kalman filter: Results with real observations. Monthly weather review, 133(3), 604-620.

Samaniego, L., Kumar, R., Attinger, S., 2010. Multiscale parameter regionalization of a grid-based hydrologic model at the mesoscale. Water Resources Research 46. doi:10.1029/2008WR007327

---

## Referee Comment (RC2) · E. Zehe (Referee) · 22 Dec 2016

Please find my comments in the attached pdf.

Best regards,

Erwin Zehe

Please also note the supplement to this comment:
http://www.hydrol-earth-syst-sci-discuss.net/hess-2016-557/hess-2016-557-RC2-supplement.pdf

[Figure]

General comments and evaluation: This manuscript introduces a coupled, cross compartment simulation of the water and energy cycles using the Neckar basin as a case study. I very much agree with the authors that coupled simulations of water and energy cycles are a key for a) advancing our fundamental understanding of environmental system dynamics and b) to identify and rectify deficiencies in data assimilation schemes. The scope of the manuscript is hence highly suited for the audience of HESS and I think that the proposed coupled model bears a huge scientific potential.

Unfortunately, the implementation of the coupled model study and its scientific presentation in the manuscript are far below the quality standard required for a publication in HESS. In the present form the paper has no clear scientific objectives. Page 3 of the introduction reads very much a like project proposal which lists all possible advantages of virtual realities – yet the manuscript does not address a single of these possible scientific objectives. This is a missed opportunity! Instead the authors provide hand waiving arguments, that plausibility of virtual simulations results is sufficient to use the virtual reality for scientific learning. I think this is a) wrong (see major point below) and b) implies that the manuscript is not reviewable, simply because plausibility of model results is nothing that can be falsified based on the provided model evidence (if the authors have a different opinion, they need to explain how to measure plausibility in an objective sense). In consequence the manuscript presents a large set of diverse and possibly very interesting simulation in results in a manner, which does not support a targeted scientific learning process beyond the fact the model may provide those results in a form that is in accordance with the mind setting of the authors.

Given the huge potential of the coupled model I strongly encourage the authors to re-submit a much more focused study, particularly with clear scientific objectives. I hope that the points listed below will be helpful for this. I have doubts whether this can be achieved within the period usually granted for major revisions, particularly also because the revision requires additional sensitivity tests with the model system.

Major points.

- In contrary to the authors' statement, I think that virtual realities are only suitable for scientific learning, if they portray non-linear systems dynamics and its sensitivity to meaningful changes in environmental characteristics in an acceptable manner. This needs to be tested using predefined evaluation criteria and acceptance thresholds, thereby avoiding bias correction, to avoid that we find what we wish to find. Data assimilation procedures which work well in an error-prone virtual reality, must not necessarily do a good job in reality, particularly not if the model is biased! A revised study could hence focus on the question whether the proposed model system performs already good enough to act as virtual reality, thereby exploring related model sensitivities. Even if this will be not the case yet, the study would be extremely interesting and valuable. Computational expense is not really a bottle neck here, as there are suitable methods to assess sensitivity of also of computational very expensive models within less than 50 runs. Another possible objective could be to quantify how much skill in water balance simulations stems from the fact that we usually drive the SVAT part of hydrological models with observed dependent data of air temperature and air humidity. In the coupled model this equivalent to the case of perfect predictions of T and air humidity in the reference layers.
- The referencing is absolutely inappropriate. The authors should acknowledge past work of competing groups in the area of coupled, cross compartment modelling, of water in energy

**Fig. 1.**

**Supplement:**

General comments and evaluation: This manuscript introduces a coupled, cross compartment simulation of the water and energy cycles using the Neckar basin as a case study. I very much agree with the authors that coupled simulations of water and energy cycles are a key for a) advancing our fundamental understanding of environmental system dynamics and b) to identify and rectify deficiencies in data assimilation schemes. The scope of the manuscript is hence highly suited for the audience of HESS and I think that the proposed coupled model bears a huge scientific potential.

Unfortunately, the implementation of the coupled model study and its scientific presentation in the manuscript are far below the quality standard required for a publication in HESS. In the present form the paper has no clear scientific objectives. Page 3 of the introduction reads very much a like project proposal which lists all possible advantages of virtual realities – yet the manuscript does not address a single of these possible scientific objectives. This is a missed opportunity! Instead the authors provide hand waiving arguments, that plausibility of virtual simulations results is sufficient to use the virtual reality for scientific learning. I think this is a) wrong (see major point below) and b) implies that the manuscript is not reviewable, simply because plausibility of model results is nothing that can be falsified based on the provided model evidence (if the authors have a different opinion, they need to explain how to measure plausibility in an objective sense). In consequence the manuscript presents a large set of diverse and possibly very interesting simulation in results in a manner, which does not support a targeted scientific learning process beyond the fact the model may provide those results in a form that is in accordance with the mind setting of the authors.

Given the huge potential of the coupled model I strongly encourage the authors to re-submit a much more focused study, particularly with clear scientific objectives. I hope that the points listed below will be helpful for this. I have doubts whether this can be achieved within the period usually granted for major revisions, particularly also because the revision requires additional sensitivity tests with the model system.

Major points.

- In contrary to the authors' statement, I think that virtual realities are only suitable for scientific learning, if they portray non-linear systems dynamics and its sensitivity to meaningful changes in environmental characteristics in an acceptable manner. This needs to be tested using predefined evaluation criteria and acceptance thresholds, thereby avoiding bias correction, to avoid that we find what we wish to find. Data assimilation procedures which work well in an error-prone virtual reality, must not necessarily do a good job in reality, particularly not if the model is biased! A revised study could hence focus on the question whether the proposed model system performs already good enough to act as virtual reality, thereby exploring related model sensitivities. Even if this will be not the case yet, the study would be extremely interesting and valuable. Computational expense is not really a bottle neck here, as there are suitable methods to assess sensitivity of also of computational very expensive models within less than 50 runs. Another possible objective could be to quantify how much skill in water balance simulations stems from the fact that we usually drive the SVAT part of hydrological models with observed dependent data of air temperature and air humidity. In the coupled model this equivalent to the case of perfect predictions of T and air humidity in the reference layers.
- The referencing is absolutely inappropriate. The authors should acknowledge past work of competing groups in the area of coupled, cross compartment modelling, of water in energy

cycles (e.g. in Hohenheim and in Sweden) and they need to explain how their approach compares to those. The author should also check the considerable body of literature on virtual landscape studies in hydrology (e.g. by Hopp, Gerrits, Ebel and Loague and many more). This might be helpful to focus the revised manuscript on clear objectives and science questions and a more targeted evaluation of the simulation results with respect to these questions.

- I truly miss a critical reflection of whether the proposed coupling of COSMO, CLM and PARFLOW at the selected grid resolution does compromise the physical basis of the concept used to parametrize shallow turbulence, of the Richards equation and of Mannings roughness as well. This is particularly astonishing because quite a few co-authors have a very strong physical background. I do not claim that there is a simple answer to questions raised below. But the investigation of land-surface atmosphere feedbacks implies, beyond coupling of models, also to enhance the theoretical fundament for this. In any case the author should be aware that a credibility of the "physics based" model paradigm stands and falls with the way how honestly we deal with the current limitation of our theories.

  o The Richards equation relies on the assumption of local equilibrium conditions. The latter is surely not fulfilled at a grid scale of 100 by 100 m (Or et al. 2015 WRR). So what is the physical meaning of a soil moisture and a matric potential defined at this scale? How does these effective quantities to observables, particularly the effective potential to binding energy density of water in the pore space? What would be the effect on simulated root water uptake and transpiration and hence latent heat release when running PARFLOW on a 1 by 1 m grid (as recommended by Or et al. 2015), using the same approaches implemented in CLM. Is it really the average binding energy needs to be represented at this grid scale (or also minimum and maximum)?

  o Though I am a layman in turbulence and boundary layer meteorology, I remember that Monin-Obukhov theory relies on a the assumption of horizontal homogeneity and quasi steady state in the Monin Obukhov layer, because it is essentially a diagnostic approach to determine wind, humidity and temperature profiles close to the land surface. Particularly also Businger Dyer stability functions imply horizontal homogeneity. I do not think that this assumption is justified at the selected grid scale?! Particularly not with respect to the fact the length scales of topography, landuse and soil heterogeneity is smaller than 100m. So how to deal with this in the future?

  o A proper accounting for river net geometries is as important for flood routing, as preferential flow paths for subsurface flows (which are neglected as well). An adjustment of Mannings n to compensate for errors is simply unphysical (as n is related to the size of roughness elements) and implies that a parameter with a physical meaning degenerates to a fudge factor and a physical approach degenerates to a conceptual approach.

Technical details

- The authors use three pedo transfer functions to estimate their soil hydraulic functions, a short note on the scale miss match would be appropriate. These functions provide an uncertainty range for all values, why not assessing the related model sensitivity?

- As far as I know there are more than 60 different approaches for stomata conductance in the market, which one was used and how do the results depend on the choice?
- It seems that ordinary kriging has been used to regionalize soil texture and soil organic content. Please provide data on the underlying variogram (nugget to sill ratio, effective ranges etc.). Secondly it is not clear how conditional simulations were used to account for sub-grid variability and on which data this has been based. Please provide details.
- The kinematic wave approximation can be rather inappropriate for open channel flow, as the water level gradient is during events not parallel to the slope of the river bed. Why not using the diffusion wave approximation?

Best regards,

Erwin Zehe

References:

- Or, D., Lehmann, P. and Assouline, S.: Natural length scales define the range of applicability of the Richards equation for capillary flows, Water Resour. Res., 51(9), 7130–7144, doi:10.1002/2015WR017034, 2015.
- Damour, G., Simonneau, T., Cochard, H. and Urban, L.: An overview of models of stomatal conductance at the leaf level., Plant. Cell Environ., 33(9), 1419–38, doi:10.1111/j.1365-3040.2010.02181.x, 2010.

---

## Author Comment (AC2) · 27 Jan 2017

Reply to reviewer 2

We would like to thank Erwin Zehe (EZ in the remainder) for the constructive comments and suggestions, which will help considerably in improving the manuscript. With regard to the general comments and evaluation, we would like to point that based on the review, the statement of the scientific objective and presentation will be improved in the revised manuscript as outlined below.

The scientific objective of the paper is to generate a virtual reality of a larger section of the terrestrial system, which can serve as a basis to do meaningful data assimilation experiments for improving model state estimations and predictions with currently available observations from the ground and from satellites. We defined four requirements for this virtual reality:
1) The model should include the dynamics of an active groundwater layer to represent possible feedback between groundwater and atmosphere;
2) We need state evolutions over several years in order to represent variability on time scales over which the compartments of the system show significant interaction.
3) The spatial extent of the simulated system must be sufficiently large to i) allow for realistic two-way interactions and cycling of water and energy between a diverse groundwater layer, the land surface and the atmosphere, and ii) to accommodate a sufficiently large number of observations with significant information content about the system (e.g. for SMOS/SMAP observations with pixel sizes of 40 km diameter we need at least of the order of 50 to 100 pixels to be able to represent and exploit the spatial variability). Thus several hundreds of kilometers need to be covered by the system.
4) The last requirement is the capacity of current HPC environments to actually perform the simulations.

Given these requirements, we set up a virtual reality with a model as close to physics as possible (the degree of which is of course debatable and is not meant in the sense of first principle), which practically also reflects the highest achievable spatial resolution in order to represent the variability at the scale of observations as best as possible. Using one of the most effective HPC environments led to an atmospheric resolution of about 1 km and a land resolution of 100 m as the highest resolution achievable today. At this point, the virtual reality cannot be resolved at 100m resolution, because of technical constraints of the used land surface model (CLM3.5), but EZ's criticism toward insufficient physics in the land models would not change by increasing resolution from 1 km to 100 m. If we needed to resolve the eddies of the ABL in order to be able to abandon the Monin-Obukhov similarity theory (MOST), we would need to go to 10 m or below in the atmosphere, which would result in days of wall clock time for one real day simulated. In that sense, we also consider Richards equation merely as a parameterization, which – that we do require from parameterizations – reproduce reality sufficiently for our purpose. The goal of the paper is to show, that given all these requirements and constraints, our results are reasonable and sufficiently mimic the dynamics and the most important observations we have of the system in order to use these observations for state improvements and predictions.

We agree with EZ and also strongly believe that the plausibility of model results needs to be demonstrated with regard to what is expected from real-world observations. As a matter of fact, presenting the results from the virtual reality without such a test would constitute a scientific proposition that cannot be falsified. For example, the term *bias*, in

the context of the plausibility check used throughout the manuscript, does not reflect a model structural error requiring bias correction, because the virtual reality is only directed at real catchment and does not require matching reality in all aspects. Indeed non-linear dynamics are reproduced very well and, thus, indeed the virtual reality is suitable for scientific learning.

In the following we address the specific points raised by EZ. We will include the original question as reference in *italics*.

 Major points.

*In contrary to the authors' statement, I think that virtual realities are only suitable for scientific learning, if they portray non-linear systems dynamics and its sensitivity to meaningful changes in environmental characteristics in an acceptable manner. This needs to be tested using predefined evaluation criteria and acceptance thresholds, thereby avoiding bias correction, to avoid that we find what we wish to find. Data assimilation procedures which work well in an error-prone virtual reality, must not necessarily do a good job in reality, particularly not if the model is biased! A revised study could hence focus on the question whether the proposed model system performs already good enough to act as virtual reality, thereby exploring related model sensitivities. Even if this will be not the case yet, the study would be extremely interesting and valuable. Computational expense is not really a bottleneck here, as there are suitable methods to assess sensitivity of also of computational very expensive models within less than 50 runs. Another possible objective could be to quantify how much skill in water balance simulations stems from the fact that we usually drive the SVAT part of hydrological models with observed dependent data of air temperature and air humidity. In the coupled model this equivalent to the case of perfect predictions of T and air humidity in the reference layers.*

The scope of the study is to represent the scientific and technical challenges in setting up a virtual reality and demonstrate plausibility. This is not a small feat, and for the first time, this has been done from groundwater across the land surface into the atmosphere. Because the system is able to reproduce highly non-linear feedbacks across different time and space scales, the model will be a valuable tool in future for e.g. sensitivity analyses. Yet, this is clearly beyond the point of the presented study. As aforementioned the system is not designed and setup to reproduce reality. Therefore, the term bias does not reflect a model structural error, but only indicates where the model does not agree systematically in terms of absolute values with reality.

*The referencing is absolutely inappropriate. The authors should acknowledge past work of competing groups in the area of coupled, cross compartment modelling, of water in energy cycles (e.g. in Hohenheim and in Sweden) and they need to explain how their approach compares to those. The author should also check the considerable body of literature on virtual landscape studies in hydrology (e.g. by Hopp, Gerrits, Ebel and Loague and many more). This might be helpful to focus the revised manuscript on clear objectives and science questions and a more targeted evaluation of the simulation results with respect to these questions.*

We regret that we missed important references, because we focused too much on our purpose of the virtual realities. Other groups also work on virtual realities although with different foci, and there are certainly similarities to point out and to compare, and we

will do so in the revised manuscript. However, this study is not intended to be a model showdown; individual models have been and still are part of model intercomparisons and the problems we encountered for our setup would also show up for other models. Indeed, a major intention of our paper is to inform the community about the problems arising, when using such models for the given problem.

*I truly miss a critical reflection of whether the proposed coupling of COSMO, CLM and PARFLOW at the selected grid resolution does compromise the physical basis of the concept used to parameterize shallow turbulence, of the Richards equation and of Mannings roughness as well. This is particularly astonishing because quite a few co-authors have a very strong physical background. I do not claim that there is a simple answer to questions raised below. But the investigation of land-surface atmosphere feedbacks implies, beyond coupling of models, also to enhance the theoretical fundament for this. In any case the author should be aware that a credibility of the "physics based" model paradigm stands and falls with the way, how honestly we deal with the current limitation of our theories.*

We agree with EZ that underlying simplifying assumptions used in the models and issues of spatial and temporal scale are an important part of the discussion, which will be expanded considerably in the revised manuscript. Some thoughts are also provided in relation to EZ's specific comments below.

*The Richards equation relies on the assumption of local equilibrium conditions. The latter is surely not fulfilled at a grid scale of 100 by 100 m (Or et al. 2015 WRR). So what is the physical meaning of a soil moisture and a matric potential defined at this scale? How does these effective quantities to observables, particularly the effective potential to binding energy density of water in the pore space? What would be the effect on simulated root water uptake and transpiration and hence latent heat release when running PARFLOW on a 1 by 1 m grid (as recommended by Or et al. 2015), using the same approaches implemented in CLM. Is it really the average binding energy needs to be represented at this grid scale (or also minimum and maximum)?*

These are very important points, which are under permanent discussion in the literature. EZ correctly identifies a critical assumption made by applying the Richards equation for such coarse grid cells. Yet, it is important to stress that this is common practice in atmospheric models and land surface models, where often (modified variants of) the Richards equation is applied on even coarser scales. It is our aim in this project to be able to apply the Richards equation at a much finer resolution, but we agree that a resolution of 100 m is still critical. Notice, however that it is exactly our aim to relax this critical point somewhat.

It is very clear that the Richards equation on a 100 m by 100 m grid cell cannot be considered a model based on first principles, and in particular lateral fluxes will not be reproduced well. We would, however, like to stress that the vertical resolution of the model is in the cm range, thus the grid resolution in the main (vertical) flow direction is not unreasonable. The assumption of flow being mainly vertical is to the best of our knowledge reasonable, except for hillslope flow. Local equilibrium in horizontal direction might not be fulfilled, however, the equilibration will not happen by lateral flow over 100 meters, thus its reproduction is limited. One should also consider that in German climatic conditions, flow in the unsaturated zone is usually slow and water content changes mostly very little, slowly, and only in the very uppermost part of the soil

(except for hillslopes, as mentioned above, and rare extreme events). A 100 m pixel can thus be considered as an average over different columns that run in parallel. If the horizontal average over these columns can be reproduced by a Richards equation could of course be debated.

We also agree that a pressure head on a 100 m scale as a variable is questionable, such as pressure heads in any hydrosystem described on this length scale (aquifers). It can only be considered an auxiliary quantity. As mass is an extensive quantity, the water content is not as questionable: This would be the spatially averaged water content over this length scale.

As a side remark about the Richards equation in general: The Richards equation relies on very strong assumptions, the assumption of local equilibrium being only one of them. As a matter of fact, the Richards equation is "wrong" down to any length scale, even to the cm scale, and the problem of lack of local equilibrium reaches down to the small length scales. Comparing pore-scale observations and simulations to Richards equations models demonstrates this. One could list many flow phenomena that are not captured by the Richards equation.
Trying to reproduce, for example, preferential flow (that is found in field tests, no question about that) with a Richards equation with heterogeneous parameters even on a cm scale requires parameter contrasts that are found in fractured rock, but not in soils. New model concepts are needed to predict preferential flow, but this lack will not be solved by using a Richards model with a finer grid resolution.

Although we agree that the Richards equation on 100 m grid cells has to be considered a simplified model, that is not expected to capture all flow phenomena correctly, we still think that it is better suited than a bucket approach (as the other extreme). First, as outlined above, the vertical resolution is reasonable. Second, the Richards equation contains the two essentials that are needed also for the large scale: A storage and a hydraulic conductivity that decreases with water content.

We should have included a paragraph on the use of the Richards equation on large length scales and should have clarified that we do not claim that we will be able to capture all flow phenomena correctly and also that we are aware that parameters for the Richards equation on the large length scales will for this reason depend on boundary conditions. The model should in this sense be considered a grey box. This makes the requirement for data assimilation stronger.

*Though I am a layman in turbulence and boundary layer meteorology, I remember that Monin-Obukhov theory relies on a the assumption of horizontal homogeneity and quasi steady state in the Monin Obukhov layer, because it is essentially a diagnostic approach to determine wind, humidity and temperature profiles close to the land surface. Particularly also Businger Dyer stability functions imply horizontal homogeneity. I do not think that this assumption is justified at the selected gridscale?! Particularly not with respect to the fact the length scales of topography, landuse and soil heterogeneity is smaller than 100m. So how to deal with this in the future?*

It is indeed correct that below the km scale, the parameterizations that we use in our COSMO version (and take up also in its coupling with CLM) start to conflict with the

resolved atmospheric dynamics (so-called grey zone). Thus, parameterizations need to become grid size-dependent, which is still an unresolved and debated topic, and authors of this paper are actually working on this topic by introducing a scale-dependent asymptotic length scale. This is also the reason, that, for now, we do not intend to increase resolution beyond 1.1 km in the atmosphere. We use instead an atmospheric downscaling to the resolution of the land models by Schomburg et al. (2010, 2012) and Zerenner et al. (2016) to account for the scale difference. However, as already said above, it would just not be possible to run COSMO – or any atmospheric model – on grid resolutions which allow to abandon MOST for our particular purpose.

*A proper accounting for river net geometries is as important for flood routing, as preferential flow paths for subsurface flows (which are neglected as well). An adjustment of Mannings n to compensate for errors is simply unphysical (as n is related to the size of roughness elements) and implies that a parameter with a physical meaning degenerates to a fudge factor and a physical approach degenerates to a conceptual approach.*

Except for the timing of river hydrographs the ParFlow already produces remarkably good results without tuning. The main problem we are confronted with is the unrealistic river widths dictated by the model resolution, which we currently cannot change. Of course, the kinematic wave approach is a parameterization, and as all parameterizations it is meant to mimic nature. As a matter of fact, in the virtual reality we only honor overland flow without channel flow. In order to be able to compare to the discharge observations the width of the channel is adjusted via Mannings n in an observation operator approach. Of course we would have to move to another concept when we attempt to assimilate river stage, which will become available from high-resolution active satellite sensors.

Preferential channels were introduced in the form of river alluvium in this virtual reality. At this point, interbedded paleochannels in fine-grained flood plain deposits as they often occur in real-world settings are not incorporated. We are aware of the importance and existence of those channels, as can also been seen from the publication record of some of the authors of this paper. However, compared to existing atmospheric models and land surface models the created virtual reality is already a big step forward by reproducing detailed spatial variability of subsurface properties. It was beyond the scope of this virtual reality to introduce more preferential flow, but this is something, which is planned for more advanced future versions of this virtual reality. The limitation will be acknowledged in the extended discussion.

Technical details

*The authors use three pedotransfer functions to estimate their soil hydraulic functions, a short note on the scale miss match would be appropriate. These functions provide an uncertainty range for all values, why not assessing the related model sensitivity?*

We briefly discuss some limitations of the soil parameterization in the current manuscript at P17/L12-23. But we agree with EZ, that also the scale mismatch is a relevant topic, which affects all hydrological applications at relatively large domains for which direct soil measurements are not available. For this reason we will discuss in the revised manuscript also this limitation. We did not perform the related model sensitivity since the aim was to generate and analyze a reference case. Based on this, we aim to

create different ensembles (i.e., by perturbing different sources of uncertainty) that will be used to evaluate the sensitivity of the model response and test data assimilation approaches.

*As far as I know there are more than 60 different approaches for stomata conductance in the market, which one was used and how do the results depend on the choice?*

There are basically two conceptually different approaches with many variants, e.g. concerning its dependency on soil moisture. We have chosen the standard implementation in CLM3.5 (the CLM manual references Collatz et al. , 1991 and Sellers et al. 1992), which we, as many other groups, extensively tested and also improved. Of course the results will change, when we use different approaches, but the current state of science does not yet allow for selecting one as the truth. We have to accept this condition and errors as uncertainties, when we perform data assimilation. In the revised version of the manuscript we will add additional information regarding the specific modeling approach used in this study.

I*t seems that ordinary kriging has been used to regionalize soil texture and soil organic content. Please provide data on the underlying variogram (nugget to sill ratio, effective ranges etc.). Secondly it is not clear how conditional simulations were used to account for sub-grid variability and on which data this has been based. Please provide details.*

The resolution of the soil map available for the entire domain (BUEK 1,000,000) is too coarse for representing variability at the model grid resolution (100 x 100 m$^2$). After a comparison of different methods available in literature for disaggregating/downscaling the information of the original soil map (Goovaerts, 2010; Heuvelink et al., 2016; Kerry et al., 2012; Ranney et al., 2015), a new and relatively simple method was developed to preserve the main spatial patterns in the original soil map while introducing sub-scale variability. Additional information regarding the approach is provided below and a schematic is provided in Figure 1, as example. Accordingly, we will provide additional information and references to clarify the approach developed also in the revised version of the manuscript.

The original soil map is defined in the example below to be a horizontal transect with three soil units with different sand [%]. Based on that:

1. A random sampling design is used to distribute point locations within the transect (blue points in Step C1). Values of sand [%] are attributed to each point location.
2. These extracted values are used to calculate experimental variograms and cross-variograms. Exponential + nugget models are fitted to all spatial structures based on least square residual as presented by (Pebesma, 2004). An example of fitting results for the first soil layer is provided in the table below. It has to be noted that additional refinements could have been conducted (e.g., testing the model fit using additional nested model), but since the aim was not to find the best spatial structure but to introduce small-scale variability, the variogram models were considered suitable for the specific application.

|          | Model  | partial sill | Range [m] |
|----------|--------|--------------|-----------|
| Sand [1] | Nugget | 20           | -         |

| | | | |
|---|---|---|---|
| Sand [2] | Exponential | 250 | 8250 |
| Clay [1] | Nugget | 6 | - |
| Clay [2] | Exponential | 275 | 3950 |
| Sand.Clay [1] | Nugget | -9 | - |
| Sand.Clay [2] | Exponential | -162 | 5450 |

3. The sample locations (blue points in the sketch) are used as conditional points for further interpolation (step C3). Interpolation by ordinary kriging tends to smooth the spatial variability. This approach was used for interpolating horizon depths and carbon content. To avoid this effect by providing sub-scale variability (<1 km$^2$) for the texture, sand and clay percentage are interpolated based on conditional co-simulation (Gómez-Hernández and Journel, 1993). In this case, the points are still used to condition the interpolation but now equally likely random fields are generated. Conditional geostatistic simulation is an approach commonly used in subsurface hydrology and we refer to specific references for additional information (Deutsch and Journel, 1998; Goovaerts, 1997; Isaaks and Srivastava, 1989).

[Figure]

Figure 1: simple sketch representing the approach used to introduce small-scale variability in the original soil map, here represented by a horizontal transect of sand [%].

*The kinematic wave approximation can be rather inappropriate for open channel flow, as the water level gradient is during events not parallel to the slope of the river bed. Why not using the diffusion wave approximation?*

While the diffusive wave approximation may yield better results during flood events, it seems so far that the kinematic wave approximation does work except in a few cases. Häfliger et al. (2015) show for example that the kinematic wave approximation works in average better than the diffusive wave in the Garonne basin (south-western France): the size of this basin (~56 000 km$^2$) is similar to the size of the Neckar catchment. Discharge validations were performed with river gauges in the river network of the catchment: the Nash-Criterium (Nash and Sutcliffe 1970) values using the kinematic wave were slightly better than the diffusive wave for several gauges in the downstream Garonne river. The used model to simulate discharge and water levels was a regional distributed model relatively similar to the ParFlow model, running with a spatial resolution of 1 km and a timestep of 300 s to simulate overland flow in the river network (RAPID river routing model, David et al. 2011a,b). Furthermore, implementing the diffusive wave into ParFlow would require considerable work and not resolve the issue we currently face with the true river width not matching the model grid size.

Hopefully we could address most of your concern.

Kind regards,

The Authors.

References

Collatz, G. J., Ball, J. T., Grivet, C., & Berry, J. A., 1991. Physiological and environmental regulation of stomatal conductance, photosynthesis and transpiration: a model that includes a laminar boundary layer. Agricultural and Forest Meteorology, 54(2-4), 107-136.

Cosby, B.J., Hornberger, G.M., Clapp, R.B., Ginn, T.R., 1984. A Statistical Exploration of the Relationships of Soil Moisture Characteristics to the Physical Properties of Soils. Water Resour. Res. 20, 682–690. doi:10.1029/WR020i006p00682

David, C.H., Habets, F., Maidment, D.R., Yang, Z.-L., 2011a. RAPID applied to the SIM-France model. Hydrol. Processes 25, 3412–3425. doi:10.1002/hyp.8070

David, C.H., Maidment, D.R., Niu, G.-Y., Yang, Z.-L., Habets, F., Eijkhout, V., 2011b: River network routing on the NHDPlus dataset. J. Hydrometeor. 12, 913–934. doi: 10.1175/2011JHM1345.1

Deutsch, C.V., Journel, A.G., 1998. GSLIB: Geostatistical Software Library and User's Guide. Oxford University Press.

Gómez-Hernández, J.J., Journel, A.G., 1993. Joint Sequential Simulation of MultiGaussian Fields, in: Soares, A. (Ed.), Geostatistics Tróia '92, Quantitative Geology and Geostatistics. Springer Netherlands, pp. 85–94. doi:10.1007/978-94-011-1739-5_8

Goovaerts, P., 2010. Combining Areal and Point Data in Geostatistical Interpolation: Applications to Soil Science and Medical Geography. Math. Geosci. 42, 535–554. doi:10.1007/s11004-010-9286-5

Goovaerts, P., 1997. Geostatistics for Natural Resources Evaluation. Oxford University Press.

Häfliger, V., Martin, E., Boone, A., Habets, F., David, C.H., Garambois, P.-A., Roux, H., Ricci, S., Berthon, L., Thévenin, A., Biancamaria, S., 2015. Evaluation of Regional-Scale River Depth Simulations Using Various Routing Schemes within a Hydrometeorological Modeling Framework for the Preparation of the SWOT Mission. Journal of Hydrometeorology 16, 1821-1842.

Heuvelink, G.B.M., Kros, J., Reinds, G.J., De Vries, W., 2016. Geostatistical prediction and simulation of European soil property maps. Geoderma Reg. 7, 201–215. doi:10.1016/j.geodrs.2016.04.002

Isaaks, E.H., Srivastava, R.M., 1989. An Introduction to Applied Geostatistics. Oxford University Press.

Kerry, R., Goovaerts, P., Rawlins, B.G., Marchant, B.P., 2012. Disaggregation of legacy soil data using area to point kriging for mapping soil organic carbon at the regional scale. Geoderma 170, 347–358. doi:10.1016/j.geoderma.2011.10.007

Nash, J.E., Sutcliffe, J.V., 1970. River flow forecasting through conceptual models part I—A discussion of principles. J. Hydrol. 10, 282–290, doi:10.1016/0022-1694(70)90255-6

Pebesma, E.J., 2004. Multivariable geostatistics in S: the gstat package. Comput. Geosci. 30, 683–691. doi:10.1016/j.cageo.2004.03.012

Ranney, K.J., Niemann, J.D., Lehman, B.M., Green, T.R., Jones, A.S., 2015. A method to downscale soil moisture to fine resolutions using topographic, vegetation, and soil data. Adv. Water Resour. 76, 81–96. doi:10.1016/j.advwatres.2014.12.003

Rawls, W.J., 1983. Estimating soil bulk density from particle size analysis and organic matter content. Soil Sci. 135.

Schomburg, A., V. Venema, R. Lindau, F. Ament, and C. Simmer, 2010: A downscaling scheme for atmospheric variables to drive soil-vegetation-atmosphere transfer models. Tellus, 62, 4, p.242-258, doi 10.1111/j.1600-0889.2010.00466.x.

Schomburg, A.,V. Venema, R. Lindau, F. Ament, and C. Simmer, 2012: Disaggregation of screen-level variables in a numerical weather prediction model with an explicit simulation of subgrid-scale land-surface heterogeneity. Meteorology and Atmospheric Physics, doi: 10.1007/s00703-012-0183-y, 116, no. 3-4, pp. 81-94.

Sellers, P. J., Berry, J. A., Collatz, G. J., Field, C. B., & Hall, F. G., 1992. Canopy reflectance, photosynthesis, and transpiration. III. A reanalysis using improved leaf models and a new canopy integration scheme. Remote sensing of environment, 42(3), 187-216.

Tietje, O., Hennings, V., 1996. Accuracy of the saturated hydraulic conductivity prediction by pedo-transfer functions compared to the variability within FAO textural classes. Geoderma 69, 71–84. doi:10.1016/0016-7061(95)00050-X

Tóth, B., Weynants, M., Nemes, A., Makó, A., Bilas, G., Tóth, G., 2015. New generation of hydraulic pedotransfer functions for Europe: New hydraulic pedotransfer functions for Europe. Eur. J. Soil Sci. 66, 226–238. doi:10.1111/ejss.12192

Zerenner, T., V. Venema, P. Friederichs, and C. Simmer, 2016: Downscaling near-surface atmospheric fields with multi-objective Genetic Programming. Environmental Modelling & Software, 84, p.85–98, DOI: http://dx.doi.org/10.1016/ j.envsoft.2016.06.009

---

## Referee Comment (RC3) · Anonymous Referee #3 · 23 Feb 2017

This paper discusses the use of the COSMO-CLM LAM together with PARFLOW within the TerrSysMP platform to simulate catchment characteristics as a virtual reality. The study contributes to the idea of providing downscaled model data for assimilation, particularly for those compartments where there are typically no observations available. This dearth of measurements makes this an interesting and important work. Regarding the modelling section, initialization is very important for good representation of land surface atmosphere feedbacks, particularly in a 'climate' mode downscaling with no data assimilation. In this regard the authors have considered the land and subsurface parameters carefully, and have run a reasonable spin-up of the simulation for soil moisture, aspects which are often poor or missing in simulations. The approach also

presupposes that important process are well captured by the model, particularly in a statistical sense for those variables where a downscaling is not expected to reproduce actual distributions, such as precipitation. But the authors take great care to explain these issues, and justify their approach. Consequently I am quite satisfied with this paper, and only have specific comments on structural points. 1. I am missing a short section at the start of the modelling methods where you lay out the complete model chain, all domains on one fig, vertical levels, parameterisations used, simulation duration, and integration time. At the moment it is spread throughout the text, which drifts rather between details on each domain, with elements about CLM and PARFLOW coming in between. This makes it difficult for modellers, particularly those unfamiliar with COSMO, to see what has been done. This is especially confusing because you appear to be running Parflow in an offline mode driven without feedback – is that correct? Perhaps you can make this clearer. You could add a single paragraph for the model chain and simulation at the beginning before then going into detail on each component, without adding too much text. This small repetition will be appreciated by modellers, particularly those who would like to cite the paper purely for justifying their own model configuration – quite common. 2. To me the authors' use of the term Virtual Reality, sounds almost like it should have a Trademark (TM) after it, and suggests that something completely new is being done by running a catchment simulation – which we know is simply not true. I would suggest at least not mentioning this term as much. You can also use the words 'downscaling' and 'simulation' too. In summary though, I suggest that this is a well-structured paper and recommend its publication with these minor revisions.

---

## Author Comment (AC3) · 14 Mar 2017

Dear Reviewer,

Thank you very much for your feedback. We are pleased that you like our work.

Regarding your two suggestions we are confident that we can address these issues in a revised version:

*I am missing a short section at the start of the modelling methods where you lay out the complete model chain, all domains on one fig, vertical levels, parameterisations used, simulation duration, and integration time. [...]*

1. Concerning the overview of the experiment and model setups: we have already planned, given the feedback from the other reviewers that we got so far, that it is not entirely clear how the different domains are set up, how the models have been used and we think it is very important to show exactly what has been done and why. We will condense all the information which is currently too much scattered throughout the manuscript.

*To me the authors' use of the term Virtual Reality, sounds almost like it should have a Trademark (TM) after it, and suggests that something completely new is being done by running a catchment simulation – which we know is simply not true. I would suggest at least not mentioning this term as much.*

2. The overuse of the term "Virtual Reality" is certainly obvious and can be corrected easily. In most cases it is not necessary and "simulation" will indeed be sufficient.

With kind regards,

The Authors